# Dutch tenants at risk of eviction: Identifying predictors of eviction orders

**Marieke H. Edwards**[1¤], **Linda van den Dries**[1], **Manfred te Grotenhuis**[2†], **Sten-Åke Stenberg**[3], **Judith R. L. M. Wolf**[1] *

**1** Impuls—Netherlands Center for Social Care Research, Radboud Institute for Health Sciences, Radboud University Medical Center, Nijmegen, The Netherlands, **2** Faculteit der Sociale Wetenschappen, Radboud Universiteit Nijmegen, Nijmegen, The Netherlands, **3** Swedish Institute for Social Research, Stockholm University, Stockholm, Sweden

† Deceased.
¤ Current address: Idaho Council on Developmental Disabilities, State of Idaho, Boise, Idaho, United States of America
* judith.wolf@radboudumc.nl

**Data Availability Statement:** The SPSS data and syntax files are available from the ICPSR database (https://doi.org/10.3886/E120762V1).

**Funding:** This study was funded by ZonMw, the Netherlands Organization for Health Research and

## Abstract

In order to prevent evictions, it is important to gain more insight into factors predicting whether or not tenants receive an eviction order. In this study, ten potential risk factors for evictions were tested. Tenants who were at risk of eviction due to rent arrears in five Dutch cities were interviewed using a structured questionnaire, and six months later their housing associations were asked to provide information about the tenants' current situation. Multiple logistic regression analyses with data on 344 tenants revealed that the amount of rent arrears was a strong predictor for receiving an eviction order. Furthermore, single tenants and tenants who had already been summoned to appear in court were more likely to receive an eviction order. These results can contribute to identifying households at risk of eviction at an early stage, and to develop targeted interventions to prevent evictions.

## Introduction

Losing one's home due to an eviction, and also the mere prospect of being evicted, can have a severe impact on the lives of those involved [1]. It causes major stress and feelings of panic and shame [2], and may even lead to suicide [3]. Although the consequences of evictions are severe, data on the impact of and reasons behind evictions are scarce [4–6]. It is therefore important to gain more insight into the factors that increase the risk of eviction. In this study we explore which factors cause eviction orders among Dutch tenants who are at risk of eviction due to rent arrears.

Preventing evictions is not only important from a social and health point of view, but also from a financial perspective [5]. Akkermans and Räkers [7] estimated the cost of an eviction in the Netherlands to be between 5,000 and 9,000 euro. These costs include bailiff and litigation costs, costs for the eviction itself, unrecovered rent, costs for repairing damages to the property, and costs for removing the personal belongings of the tenants from the accommodation. When tenants are not able to pay these costs, they have to be covered by the housing

Development, grant number 204002002. The award was received by Judith R.L.M. Wolf. The ZonMw website is: https://www.zonmw.nl/en/. The funders had no role in study design, data collection and analysis, decision to publish, or preparation of the manuscript.

**Competing interests:** The authors have declared that no competing interests exist.

associations. There are also costs for society, for example, due to the use of the social relief system, and costs related to rehousing [8], debt counseling [9] and the utilization of shelters facilities.

There are approximately 350 social housing associations in the Netherlands, owning about a third of the total housing stock; they house around four million tenants in 2.4 million homes [10]. Tenants with arrears of roughly two months' rent receive a letter from their social housing association requesting them to pay the rent arrears or make payment arrangements. When the terms of the housing association are not met within approximately three months, the debt is handed over to a bailiff who then tries to collect the rent or make payment arrangements. If this fails, the bailiff requests the court to serve the tenant with an eviction order, which the housing association can use to terminate the tenancy agreement [5]. In the Netherlands, receiving an eviction order does not necessarily mean that a tenant will be evicted. In almost 75% of the cases in which tenants receive an eviction order, the social housing association and the tenant come to agreements which prevent the eviction, for example repayment in installments [11]. In 2018, an estimated 12,000 eviction orders were issued in the Dutch social housing sector, of which 3,000 orders were actually effectuated, and 1,500 tenants left the residence after receiving the eviction order, before an eviction could take place. Of the 3,000 evictions, the majority, 80%, was due to rent arrears [12].

The negative social, health and financial consequences of evictions underscore the importance of gaining insight into the predictors of eviction in order to prevent evictions and the accompanying negative effects. In a recent large study in Milwaukee [13], individual, neighborhood, and social network determinants of eviction were identified. This study showed that a higher number of children in the household, recent job loss, high neighborhood crime and eviction rates, and network disadvantage (strong ties with people who had experienced or were experiencing poverty and disadvantage) increased the likelihood of eviction. Furthermore, a survey among tenants appearing in the Milwaukee eviction court indicated that tenants with children were significantly more likely to receive an eviction order than tenants without children [14]. However, several Dutch housing associations claim to be more lenient towards households with children, because it is agreed that, if possible, eviction of children should be prevented [15]. Risk factors for eviction identified by Stenberg, Kareholt, and Carroll [16], are a low income, a criminal record, being refused help from the welfare authorities and being an immigrant. In a longitudinal study on home owners and renters in Britain from 1991 to 1997, Böheim and Taylor [17] concluded that households that were evicted tended to have younger heads, a lower household income, and more often had experienced a financial setback than households that were not evicted. Van Laere, De Wit, and Klazinga [18] studied a group of tenants with rent arrears in Amsterdam, and compared characteristics of those that were and were not evicted. Being native Dutch and having a drug-related problem were identified as risk factors for eviction. Furthermore, debt counseling can be helpful in preventing evictions [2,19]. Thus, a lack of help from debt counseling services may be a potential risk factor for eviction. Additionally, data from Dutch housing associations show that the majority of evicted tenants are single [12]. In addition to the abovementioned risk factors for evictions, it is plausible that the height of the rent arrears and the phase in the eviction process are also important predictors for evictions due to rent arrears, because higher rent arrears are more difficult to repay. The higher the rent arrears get, and the further tenants are in the eviction process, the less possibilities there are for recovery [7]. Furthermore, policies to prevent evictions differ significantly across Dutch cities [7,15], and therefore the city in which the tenant lives may affect their chance of recovering from their rent arrears and averting eviction. In most Dutch municipalities there are agreements between housing associations, social work agencies, debt counseling services, and in most cases the municipality, to prevent evictions. However, these

agreements differ significantly across municipalities; some agreements have clear guidelines, measurable goals, and clear descriptions of tasks of the different parties involved, while other agreements are more general and lack these details [7].

For the present study, we approached tenants who had received a second notification from a bailiff due to non-payment of rent. This is the last phase in the eviction process before the bailiff starts the court proceeding in order to obtain an eviction order. The aim of this study was to gain insight into the role of the abovementioned risk factors in predicting whether or not Dutch tenants at risk of eviction due to rent arrears eventually receive an eviction order. This knowledge will help housing associations and social workers to identify vulnerable households and develop early, targeted interventions to prevent evictions.

## Method

### Procedure and participants

For this study, we conducted logistic regression analyses with a sample of 344 Dutch households at risk of eviction due to rent arrears. Tenants were included in the study when they were at least 18 years old, lived in independent housing from social housing associations, and had received at least a second notification from a bailiff because of rent arrears for the housing they were currently living in. We have complied with APA ethical principles in our treatment of individuals participating in our research, and our study complies with the criteria for studies that have to be approved by an accredited Medical Research Ethics Committee. Upon consultation, the Arnhem/Nijmegen Ethics Committee stated that the study was exempt from formal approval (registration number 2011/110) as the participants were not subjected to any treatment other than the interview.

Tenants were contacted through sixteen social housing associations in five different municipalities in the Netherlands (Amsterdam, Leiden, Nijmegen, Rotterdam, and Utrecht). The social housing associations, or bailiffs working for them, screened all tenants with rent arrears to identify tenants who met the study's inclusion criteria. All identified tenants received an information letter about the study. Two types of letters were sent: opt-out and opt-in letters. With the opt-out method, tenants were informed that they would be contacted by telephone to ask them if they were willing to participate in the study, unless they sent in the opt-out card. With the opt-in method, tenants were asked to contact the researchers when they were willing to participate. The opt-in method was only used when the social housing association was unwilling or unable to use the opt-out method due to organizational or privacy reasons. In addition, seven local projects working with people at risk of eviction and six debt counseling agencies were provided with flyers explaining the research, which they distributed among clients that met our inclusion criteria. All these institutions and agencies chose the opt-in method, so their clients were asked to contact the researchers if they were willing to participate.

In total, 495 tenants were included in the study at T0 (Fig 1). All tenants were interviewed between November 2011 and February 2013, using a structured questionnaire with standardized instruments. Informed consent was obtained before the start of the interview. After participation, tenants received 20 euro. Six tenants could not be interviewed in Dutch. These interviews were conducted using an English translation of the questionnaire ($n = 3$), or with on the spot translation to French ($n = 2$) or Turkish ($n = 1$) by bilingual interviewers.

All interviewed tenants gave written permission to the researchers to obtain information about their situation from their housing association. This T1 data collection took place six months after each T0 interview. At T1, the respective housing association was asked whether or not an eviction order had been served to the tenant. This information was collected through

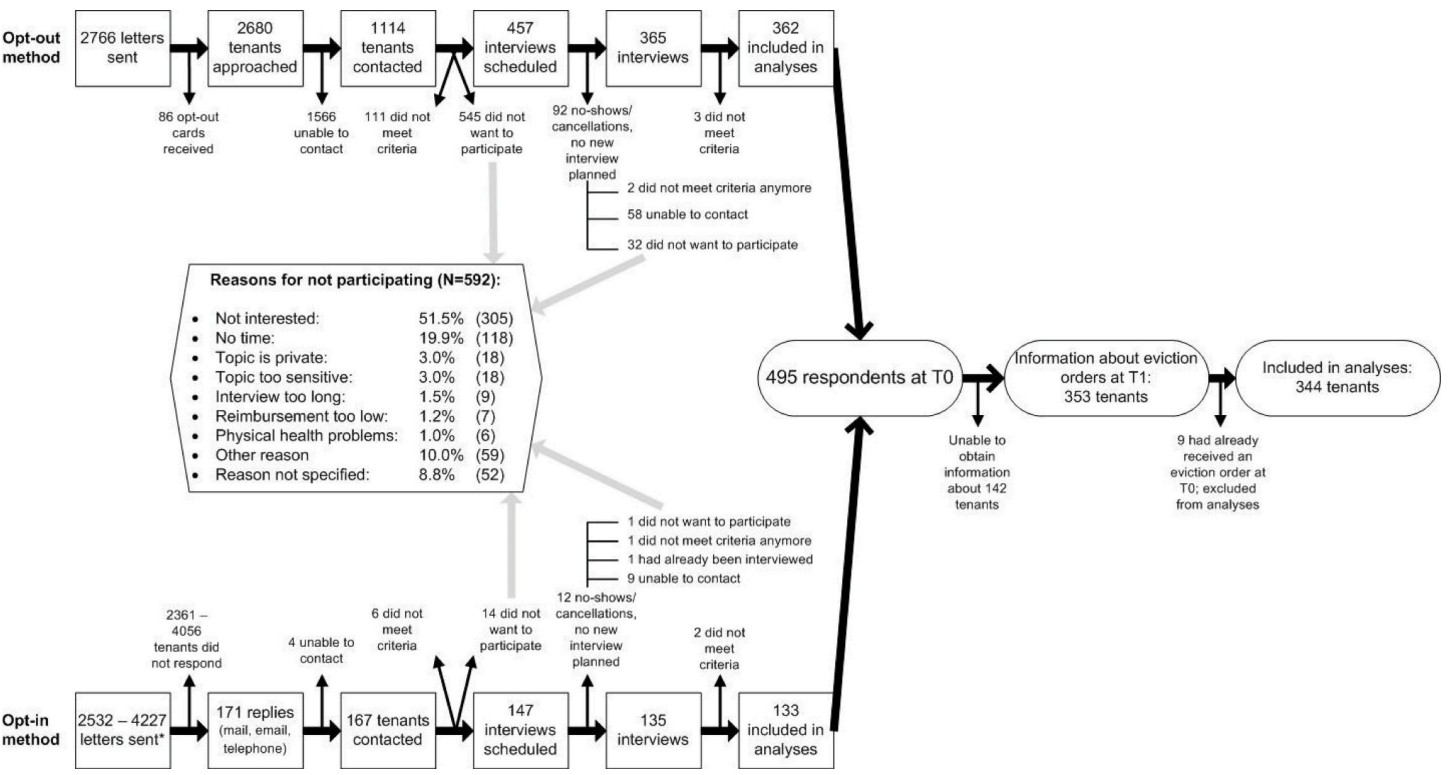

**Fig 1. Response and non-response at T0 and T1, using the opt-in and opt-out methods.** *Since not all housing associations, municipal institutions and debt counseling agencies kept track of the number of letters they sent, it is unclear exactly how many tenants received opt-in letters or flyers.

an online questionnaire for most housing associations, while two housing associations preferred to send this information by e-mail. T1 data collection took place between May 2012 and December 2013. While all participating housing associations had agreed to provide us with the T1 data, collecting this information at T1 proved to be difficult. Several housing associations were reorganizing during the course of this study, so contact persons that had agreed to provide us with the required information were no longer working at the housing association when we eventually asked for a tenant's information and their colleagues were not willing to participate. One housing association had changed their data system and was unable to find information about specific tenants. Furthermore, since several housing associations transferred rent arrear cases to debt collection agencies which then worked on these cases independently, these housing associations were unaware of the status of court proceedings. The debt collection agencies were unwilling to participate in this study, and therefore these housing associations were not able to provide information about eviction orders. We were able to obtain T1 information about eviction orders for 71% (353) of the interviewed tenants. Nine tenants had already received an eviction order at T0; these tenants were excluded from the analyses. Thus, 344 respondents were included in the analyses.

A possible bias might be caused by tenants' high levels of stress, anxiety, shame, or fear of being evicted. Consequently, we made every effort to explain that participation in the study was very important. Researchers were also very accommodating with rescheduling interview appointments when needed. We conducted several preliminary analyses to determine whether the imminent eviction was indeed perceived as an emotional burden by the tenants included in the study. We found that tenants participating in the study reported that, as a result of their rent arrears and risk of eviction, they experienced stress (87%), had trouble sleeping (66%),

were sad (72%), felt powerless (77%) and felt ashamed (57%). This suggests that tenants with a high self-assessment of eviction risk also entered the study.

## Measures

We included one outcome variable in our analyses, and ten potential predictors, derived from our review of the literature.

**Outcome variable: Eviction order.** Six months after each interview, the respective housing associations were asked whether or not the tenant had received an eviction order.

**Predictors.** Based on the literature, we included ten predictors, divided into three clusters: socio-economic variables, housing and finances, and eviction circumstances.

*Socio-economic variables.* Four socio-economic variables were included as potential predictors. The first was the respondent's age at the time of interview. Second, a foreign background variable classified tenants into three categories, based on the classification by Statistics Netherlands [20]: native Dutch (both parents were born in the Netherlands, even if the respondent was not born in the Netherlands), first generation immigrants (the respondent and at least one of the parents were not born in the Netherlands), and second generation immigrants (the respondent was born in the Netherlands and at least one of the parents was not born in the Netherlands). The household composition variable indicates whether respondents lived in one-person households or multi-person households, and the fourth variable indicates whether there were children living in the household (yes/no).

*Housing and finances.* Four variables related to housing and finances were included. Respondents estimated their total household income in the last month in Euros, and the total amount of their current rent arrears in Euros. For presentational reasons, we transformed the income and rent arrears variables by dividing them by 1,000. Furthermore, respondents were asked whether they had been fired from a job in the past three years (yes/no), and whether they had received any help from a debt counseling agency in the six months prior to the interview (yes/no).

*Eviction circumstances.* Two variables were included to account for the heterogeneity of our sample of households: the city in which the tenant lived (Amsterdam, Leiden, Nijmegen, Rotterdam or Utrecht), and the phase in the eviction process at the time of interview (before being summoned to appear in court, between summoning and the court hearing, after the court hearing but without an eviction order). The phase in the eviction process was determined by asking several questions, to be answered by yes or no (i.e. "Were you summoned to appear in court because of your rent arrears?").

## Data analysis

Table 1 shows descriptive statistics and missing values for our sample. Since data was missing for more than 5% of the respondents for income and level of rent arrears, we searched for differences between the respondents with and without values for these two variables for the outcome measure. A significant difference was found only for respondents with and without missing values for the level of rent arrears: respondents with missing values for the level of rent arrears more often received an eviction order (41%, compared to 22% of tenants with a value for level of rent arrears; $\chi^2(1, N = 344) = 5.37, p = .02$).

We used logistic regression analysis to determine which variables significantly predicted whether tenants had received an eviction order at T1. In all analyses, *p*-values of $\leq$ .10 were considered statistically significant. A backward stepwise logistic regression analysis was conducted, starting with a model that included all ten predictors and, in each step, deleting the predictor with the lowest significance, leading to a sparse model with only significant

**Table 1. Descriptive statistics of predictor variables for available responses.**

| Variable name | N | % | Missing | Mean | Range | SE |
|---|---|---|---|---|---|---|
| Age | 338 | | 6 (1.7%) | 43.4 | 19–80 | 12.3 |
| Foreign background | 343 | | 1 (0.3%) | | | |
| Native Dutch | 169 | 49.3 | | | | |
| First generation immigrant | 140 | 40.8 | | | | |
| Second generation immigrant | 34 | 9.9 | | | | |
| Household composition | 341 | | 3 (0.9%) | | | |
| One-person household | 180 | 52.8 | | | | |
| Multi-person household | 161 | 47.2 | | | | |
| Children living in the household | 341 | | 3 (0.9%) | | | |
| No | 210 | 61.6 | | | | |
| Yes | 131 | 38.4 | | | | |
| Total household income (Euro) | 321 | | 23 (6.7%) | 1405.7 | 0.0–4,500.0 | 723.5 |
| Total rent arrears (Euro) | 315 | | 29 (8.4%) | 855.3 | 0.0–5,000.0 | 941.6 |
| Fired from job in past 3 years | 335 | | 9 (2.6%) | | | |
| No | 227 | 67.8 | | | | |
| Yes | 108 | 32.2 | | | | |
| Received help from debt counseling in past 6 months | 335 | | 9 (2.6%) | | | |
| No | 253 | 75.5 | | | | |
| Yes | 82 | 24.5 | | | | |
| City | 344 | | - | | | |
| Amsterdam | 40 | 11.6 | | | | |
| Leiden | 15 | 4.4 | | | | |
| Nijmegen | 52 | 15.1 | | | | |
| Rotterdam | 62 | 18.0 | | | | |
| Utrecht | 175 | 50.9 | | | | |
| Phase in the eviction process | 334 | | 10 (2.9%) | | | |
| Before summoning | 248 | 74.3 | | | | |
| Between summoning and court hearing | 42 | 12.6 | | | | |
| After hearing, no eviction order (yet) | 44 | 13.2 | | | | |

predictors. Since only respondents with values for all predictors could be included in the backward stepwise logistic regression, the accumulation of missing values reduced our sample significantly (to $N = 275$); therefore, we followed up by adding each predictor to the sparse model individually (with the $N$ that was available for this smaller model) to determine whether adding the predictor improved the model. Using the final model, the predicted probability and marginal effects were calculated.

## Results

Of the 344 tenants that could be included in the analyses to predict eviction orders, 24% (82) had received an eviction order at T1. The initial logistic regression analysis with all predictors only shows total rent arrears at T0, living in Utrecht, and being between a summons and a court hearing as significant predictors (Table 2).

Backward stepwise logistic regression analysis resulted in a model with household composition, the level of rent arrears and phase in the eviction process as predictors. We then added each of the excluded predictors to this sparse model to determine if they improved the model, but we did not identify other significant predictors. Thus, our final model to predict eviction

**Table 2. Initial multiple logistic regression model predicting receiving an eviction order (N = 275).**

| Variable | B | SE | OR | 95% CI for OR |
|---|---|---|---|---|
| Age | 0.01 | 0.01 | 1.01 | [0.98, 1.03] |
| Foreign background (Ref: Native Dutch) | | | | |
| First generation immigrant | 0.16 | 0.38 | 1.18 | [0.56, 2.47] |
| Second generation immigrant | 0.07 | 0.64 | 1.07 | [0.30, 3.77] |
| Household composition: One-person household | 0.92 | 0.65 | 2.51 | [0.70, 8.97] |
| Children living in the household: yes | 0.04 | 0.67 | 1.04 | [0.28, 3.88] |
| Total household income/1,000 | 0.01 | 0.27 | 1.01 | [0.60, 1.71] |
| Total rent arrears at T0/1,000 | 0.94*** | 0.18 | 2.55 | [1.78, 3.64] |
| Fired from job in past 3 years: yes | 0.44 | 0.36 | 1.56 | [0.78, 3.12] |
| Received help from debt counseling in past 6 months: yes | 0.44 | 0.39 | 1.55 | [0.72, 3.34] |
| City (Ref: Rotterdam) | | | | |
| Amsterdam | 0.69 | 0.63 | 1.99 | [0.58, 6.83] |
| Leiden | 1.65* | 0.77 | 5.23 | [1.16, 23.56] |
| Nijmegen | 0.87 | 0.61 | 2.39 | [0.73, 7.86] |
| Utrecht | 0.42 | 0.54 | 1.52 | [0.53, 4.39] |
| Phase in eviction process (Ref: before summoning) | | | | |
| Between summoning and court hearing | 0.91* | 0.49 | 2.48 | [0.95, 6.46] |
| After hearing, no eviction order (yet) | 0.14 | 0.50 | 1.15 | [0.43, 3.07] |
| Constant | -4.09 | | | |

*Notes.* Reference categories are given in parentheses for categorical variables. $R^2$ = .16 (Cox & Snell) .25 (Nagelkerke).

*$p < .10$.

**$p < .001$.

***$p < .001$.

orders includes being a single tenant, total rent arrears at T0, and phase in the eviction process (Table 3).

The level of rent arrears at T0 proved to be a strong predictor for eviction orders: the odds of receiving an eviction order were more than two times greater when rent arrears were increased by € 1,000 (OR = 2.48; 95% CI = 1.79, 3.44). Furthermore, the odds of receiving an eviction order were more than two times greater for single tenants compared to households of

**Table 3. Final multiple logistic regression model predicting receiving an eviction order (N = 304).**

| Variable | B | SE | OR | 95% CI for OR |
|---|---|---|---|---|
| Household composition (Ref: multi-person household) | | | | |
| One-person household | 0.79* | 0.33 | 2.20 | [1.16, 4.16] |
| Total rent arrears at T0/1,000 | 0.91*** | 0.17 | 2.48 | [1.79, 3.44] |
| Phase in eviction process (Ref: before summoning) | | | | |
| Between summoning and court hearing | 1.17** | 0.43 | 3.22 | [1.40, 7.40] |
| After hearing, no eviction order (yet) | 0.55 | 0.42 | 1.74 | [0.76, 3.98] |
| Constant | -2.93 | | | |

*Notes.* Reference categories are given in parentheses for categorical variables. $R^2$ = .17 (Cox & Snell) .25 (Nagelkerke).

*$p < .10$.

**$p < .01$.

***$p < .001$.

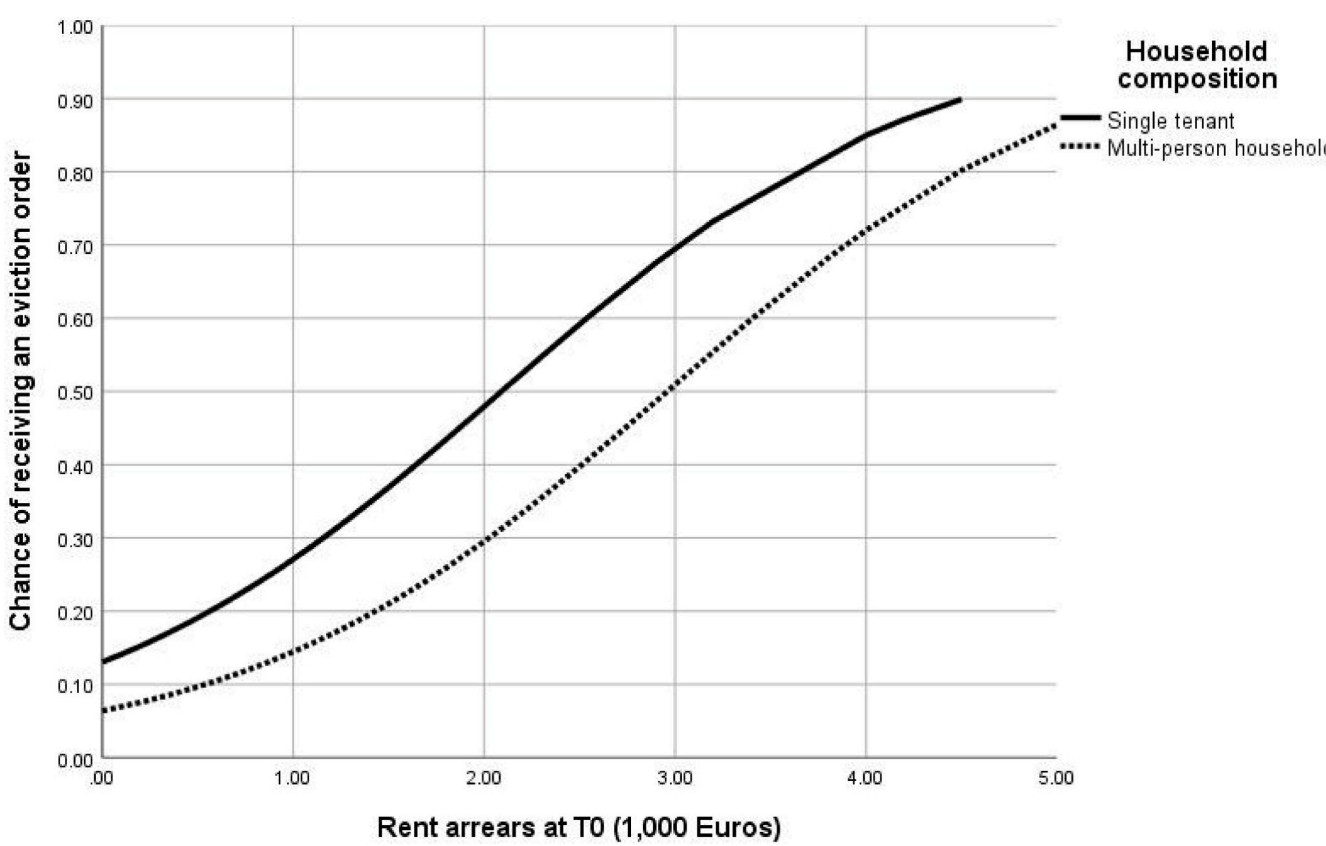

**Fig 2. Probability of receiving an eviction order as a result of level of rent arrears at T0 for different household compositions, while phase in the eviction process is at average level (N = 304).**

more than one person (*OR* = 2.20; 95% CI = 1.16, 4.16). Marginal effects indicated that, controlled for rent arrears and phase in the eviction process, single tenants had a probability of .32 (95% CI = .23, .43) of receiving an eviction order, while the probability for multi-person households was .18 (95% CI = .11, .27). Additionally, the odds of receiving an eviction order were more than three times greater for tenants who had been summoned to appear in court but had not had a court hearing yet, compared to tenants who had not been summoned to appear in court (*OR* = 3.22; 95% CI = 1.40, 7.40). Controlled for household composition and rent arrears, the probability of receiving an eviction order for tenants who had not been summoned was .15 (95% CI = .11, .21), while it was .37 (95% CI = .22, .55) for tenants who had been summoned but had not had a hearing yet, and the probability was .24 (95% CI = .13, .40) for tenants who had had a hearing.

This logistic regression model was used to calculate the probability of receiving an eviction order as a result of the level of rent arrears at T0. Figs 2 and 3 show how the probability of receiving an eviction order differs for different categories of household composition and phases in the eviction process at T0. In each of these figures, the other predictor was at average level.

In general, the probability of receiving an eviction order is relatively low for low levels of rent arrears. However, the probability of receiving an eviction order significantly increase for higher levels of rent arrears, with a higher probability for single tenants. As expected, tenants who had not yet been summoned to appear in court at T0 had the lowest probability to receive an eviction order; this probability increased with higher rent arrears. However, while tenants

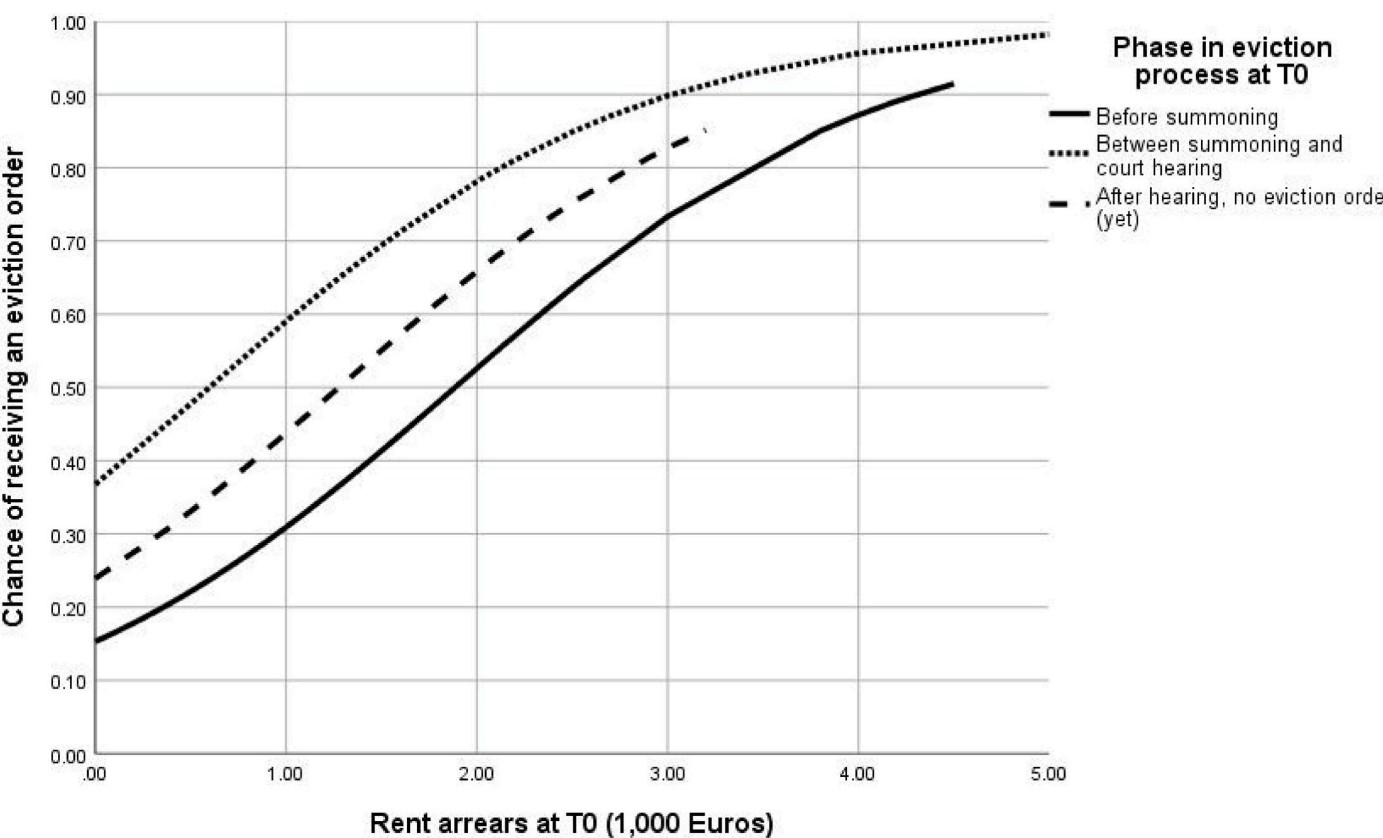

**Fig 3. Probability of receiving an eviction order as a result of level of rent arrears at T0 for different phases in the eviction process, while household composition is at average level (N = 304).**

who had been summoned and were waiting for the court hearing had the highest probability to receive an eviction order, this probability was lower (and not significantly different from the tenants who had not yet been summoned) for the tenants who had already appeared in court but had not received an eviction order (yet).

In order to further explore the data, we repeated the logistic regression analyses for single tenants and for multi-person households. Tables 4 and 5 present the final models for these groups.

**Table 4. Final multiple logistic regression model predicting receiving an eviction order for single tenants (N = 149).**

| Variable | B | SE | OR | 95% CI for OR |
|---|---|---|---|---|
| Total rent arrears at T0/1,000 | 1.10*** | 0.29 | 2.99 | [1.69, 5.30] |
| Fired from job in past 3 years: yes | 0.82* | 0.45 | 2.28 | [0.95, 5.47] |
| Phase in eviction process (Ref: before summoning) | | | | |
| Between summoning and court hearing | 2.00** | 0.68 | 7.40 | [1.96, 27.89] |
| After hearing, no eviction order (yet) | 0.46 | 0.60 | 1.58 | [0.48, 5.16] |
| Constant | -2.66 | | | |

*Notes.* Reference categories are given in parentheses for categorical variables. $R^2$ = .22 (Cox & Snell) .32 (Nagelkerke).

*$p < .10$.

**$p < .01$.

***$p < .001$.

**Table 5. Final multiple logistic regression model predicting receiving an eviction order for multi-person households (N = 151).**

| Variable | B | SE | OR | 95% CI for OR |
|---|---|---|---|---|
| Total rent arrears at T0/1,000 | 0.97*** | 0.24 | 2.65 | [1.66, 4.21] |
| City (Ref: Rotterdam) | | | | |
| Amsterdam | 2.09* | 1.00 | 8.08 | [1.14, 57.11] |
| Leiden | 2.16* | 1.17 | 8.71 | [0.88, 86.57] |
| Nijmegen | 2.18* | 1.01 | 8.81 | [1.22, 63.42] |
| Utrecht | 1.12 | 0.92 | 3.07 | [0.50, 18.81] |
| Constant | -4.04 | | | |

*Notes*. Reference categories are given in parentheses for categorical variables. $R^2$ = .18 (Cox & Snell) .30 (Nagelkerke).

*$p$ < .10.

**$p$ < .01.

***$p$ < .001.

For both single tenants and multi-person households, the level of rent arrears at T0 was a strong predictor of receiving an eviction order, but differences between these groups were found for other predictors.

For single tenants, the odds of receiving an eviction order were three times greater when rent arrears were increased by € 1,000 (*OR* = 2.99; 95% CI = 1.69, 5.30). Furthermore, the odds of receiving an eviction order were more than two times greater for single tenants who had been fired from a job in the past three years compared to single tenants who had not experienced a recent job loss (*OR* = 2.28; 95% CI = 0.95, 5.47). Additionally, the odds of receiving an eviction order were more than seven times greater for single tenants who had received a summons from a bailiff compared to single tenants who had not received a summons. Marginal effects indicated that, controlled for rent arrears and phase in the eviction process, single tenants who had recently been fired from a job had a probability of .45 (95% CI = .28, .64) of receiving an eviction order, while the probability for single tenants who had not recently lost a job was .27 (95% CI = .16, .41). Controlled for rent arrears and recent job loss, single tenants had a probability of 0.19 (95% CI = .13, .29) to receive an eviction order if they had not been summoned yet, a probability of .64 (95% CI = .34, .86) if they had received a summons but had not had a court hearing yet, and a probability of .28 (95% CI = .12, .53) if they had had a court hearing.

For multi-person households, only the level of rent arrears and the city were found to be significant predictors of receiving an eviction order. For these households, the odds of receiving an eviction order were almost three times greater when rent arrears were increased by € 1,000 (*OR* = 2.65; 95% CI = 1.66, 4.21). Furthermore, city was a significant predictor among multi-person households: compared to multi-person households in Rotterdam, the odds of receiving an eviction order were eight times greater for multi-person households in Amsterdam (*OR* = 8.08; 95% CI = 1.14, 57.11), and almost nine times greater for multi-person households in Leiden (*OR* = 8.71; 95% CI = 0.88, 86.57) and Nijmegen (*OR* = 8.81; 95% CI = 1.22, 63.42). Controlled for rent arrears, the probability for multi-person households to receive an eviction order was 0.27 (95% CI = .10, .53) in Amsterdam, .28 (95% CI = .07, .67) in Leiden, .28 (95% CI = .13, .51) in Nijmegen, .04 (95% CI = .01, .20) in Rotterdam, and .12 (95% CI = .06, .22) in Utrecht.

## Discussion

The aim of this study was to gain more insight into risk factors for receiving an eviction order. Our results indicate that the level of rent arrears is a strong predictor; higher rent arrears

significantly increase the risk of receiving an eviction order. Additionally, single tenants were more likely to receive an eviction order, and tenants who had been summoned to appear in court and were waiting for their court hearing were more likely to receive an eviction order, compared to tenants who had not been summoned yet. Furthermore, different predictors were found for single tenants and for multi-person households. While rent arrears was found to be a significant predictor in both groups, among single tenants recent job loss and phase in the eviction process were significant predictors, while among multi-person households city was a significant predictor.

Our results confirm that the level of rent arrears is an important predictor for receiving an eviction order. Increasing rent arrears decrease tenants' possibilities for recovery. Furthermore, the phase in the eviction process at the time of interview was found to be a significant predictor for eviction orders. Tenants who were between summoning and the court hearing had a higher chance to receive an eviction order than tenants who had not been summoned yet. This seems to comfirm the notion that the further tenants are in the eviction process, the smaller their chances are for recovery [7]. However, the group of tenants that was the furthest in the eviction process (the group that had already had a court hearing), did not have a significantly higher chance to receive an eviction order than tenants who had not been summoned yet. This may be explained by the fact that a court hearing can be a traumatic event for a tenant. The threat of an imminent eviction, when the housing association has the legal right to terminate the rental contract, may also serve as a strong motivator for tenants to take action towards repaying their rent arrears. Another explanation of this result is a certain selection bias: a court hearing is a very stressful event for tenants, so most tenants in that phase may not have wanted to participate in an interview. The tenants who did agree to be interviewed may have been the ones who were in less stressful circumstances, because they may have been given a second chance to repay their arrears in court, and therefore their chances of eventually receiving an eviction order were lower than for other tenants who had had a court hearing.

Furthermore, this study demonstrates that single tenants are at a significantly higher risk of receiving an eviction order. This is in line with the observations of Dutch housing associations that the majority of evicted households are single tenants [12]. A previous study [21] also demonstrated that single tenants, especially men, are at an increased risk of eviction, and a study in Sweden [22] showed that 70% of the evicted households were single tenants. The lack of support from a partner or other household members may make it more difficult for these tenants to find a solution for their rent arrears and to avert receiving an eviction order.

Further investigation of differences between single tenants and multi-person households showed significant differences between these groups. Among single tenants, besides rent arrears and phase in the eviction process, recent job loss was a significant predictor of receiving an eviction order. Single tenants generally do not have multiple income sources, so losing an income has a bigger impact than it has on families with multiple sources of income.

Interestingly, among multi-person households the city is a significant predictor of receiving an eviction order. Households in Rotterdam and Utrecht were less likely to receive an eviction order than households in Amsterdam, Leiden and Nijmegen. This indicates that different local policies and housing association procedures affect households' chances to receive an eviction order.

The results of our study have several implications for policies regarding the prevention of evictions, and provide insights that can contribute to developing targeted interventions to prevent evictions. First, since the level of rent arrears and the phase in the eviction process are strong predictors, early interventions are necessary; if at-risk households can be identified at an early stage, the rent arrears are more manageable and evictions can be prevented. Since many households with rent arrears have other debts as well, and the rent often is not the first

unpaid bill, early identification of households with a variety of arrears cannot be done by housing associations alone. In Amsterdam, for example, the *Vroeg Eropaf* ("go for it") policy, which helps to identify households with financial difficulties at an early stage, in order to find solutions before a court process starts, indeed includes a large insurance company and utility providers, besides all social housing associations and social care institutions in the city [23].

Additionally, as single tenants are at a significantly higher risk of receiving an eviction order, special attention is needed for this group. Single tenants may need extra support and professional help in order to avert eviction. Single tenants who have recently experienced a job loss have a particularly higher chance of receiving an eviction order. This knowledge may be helpful to housing associations as they develop early interventions.

It should be noted that the population of tenants at risk of eviction due to rent arrears may differ across countries, resulting from different local circumstances and policies. As this study indicated, risk factors even differ across Dutch cities; multi-person households in Rotterdam and Utrecht were less likely to receive an eviction order compared to multi-person households in Amsterdam, Leiden and Nijmegen. Each Dutch city has their own policies and projects regarding evictions. In Rotterdam, for example, intensive support is provided to families with severe problems, where families risk losing social assistance benefits or an eviction if they refuse the support that is offered [24]. While policies differ among cities, housing associations also have differing policies and procedures when families have accumulating rent arrears. Therefore, when developing interventions, it is important to study the local context and identify vulnerable households in that specific context. This study has provided insights into risk factors for eviction, and how these risk factors differ across groups. Similar studies in other countries are needed to determine if these risk factors are relevant in those contexts as well.

One of the strengths of this study is its large scale. We included 344 tenants from five municipalities of different sizes, in order to make our sample more representative of the total Dutch population of tenants at risk of eviction due to rent arrears. To our knowledge, this is the first study that specifically focused on tenants facing eviction because of rent arrears, thus providing important clues to improve policy and practice to prevent evictions of these households. This vulnerable population was difficult to contact, because many tenants did not open their mail or answer their telephone, or indicated that the anxiety and stress made it impossible for them to participate in an interview. This may have caused a selection bias at T0: tenants who were experiencing high levels of stress, anxiety and/or shame may have been less inclined to participate in an interview. However, tenants participating in the study reported high levels of stress, sadness, powerlessness, trouble sleeping, and shame; this indicates that for many tenants, these emotions did not prevent them from participating.

Another challenge related to data collection was the T1 data collection. T1 data was collected by contacting the participating housing associations and asking whether or not an eviction order was issued for the tenant (after receiving written permission from the tenant at T0 to do so). Not all housing associations were able and/or willing to provide us with the complete information six months after each interview, despite our efforts to ensure that this information would be provided. All this led to a rather high non-response for both our interview data and data from housing associations. However, due to the large scale of this study, there was still ample data to build our conclusions on. Our analyses of the missing data indicated that tenants who had a missing value for the level of rent arrears at the time of interview received an eviction order more often than tenants with a value for the level of rent arrears. It is possible that the tenants who did not answer the question about their rent arrears had lost control over and insight into their financial situation, or were unwilling to mention the level of rent arrears out of shame for the height of their debt. Therefore, our results may have underestimated the true effect of the level of rent arrears as a predictor. Another limitation of this study is that it was

impossible to predict actual evictions. Because the process from rent arrears to eviction can be very long, it was not feasible to determine which tenants were eventually evicted, as this usually takes longer than six months. Eviction orders are often used by housing associations to pressure tenants into accepting help from debt counseling. The threat of an imminent eviction, when the housing association has the legal right to terminate the rental contract, may also serve as a strong motivator for tenants to take action towards repaying their rent arrears. Therefore, there may be a long process after an eviction order, and if this process eventually leads to an eviction, it often does not take place within a few months after the eviction order.

While this study has provided some important insights into the risk factors for eviction orders, there is a great need for future research, to gain more insight into this vulnerable population and to develop targeted interventions. First, similar research should be conducted in other countries, in order to determine whether risk factors are similar across countries. Second, longer-term studies are needed to determine which risk factors are associated with actual evictions. Furthermore, it is important to examine why single tenants are at a higher risk of receiving an eviction order. More insight into all of the above will help to develop targeted, effective interventions to prevent evictions.

To summarize, this study aimed to identify risk factors for tenants at risk of eviction due to rent arrears. Our results call for early identification of households with financial difficulties, because higher rent arrears and being later in the eviction process make recovery more difficult. Single tenants are at a higher risk to receive an eviction order. Therefore, targeted interventions should be developed to take these risks into account.

## Supporting information

**S1 File. Vragenlijst predictiestudie DEF maart 2012.**
(PDF)

## Acknowledgments

We would like to thank all the tenants who took the time to participate in this study, all the interviewers, and all the housing associations who helped us to contact tenants. Manfred te Grotenhuis passed away before the submission of the final version of this manuscript. Judith Wolf accepts responsibility for the integrity and validity of the data collected and analyzed.

## Author Contributions

**Conceptualization:** Marieke H. Edwards, Linda van den Dries, Judith R. L. M. Wolf.

**Data curation:** Marieke H. Edwards, Linda van den Dries.

**Formal analysis:** Marieke H. Edwards, Manfred te Grotenhuis.

**Funding acquisition:** Judith R. L. M. Wolf.

**Investigation:** Marieke H. Edwards.

**Methodology:** Marieke H. Edwards, Judith R. L. M. Wolf.

**Project administration:** Linda van den Dries.

**Supervision:** Linda van den Dries, Judith R. L. M. Wolf.

**Visualization:** Marieke H. Edwards, Manfred te Grotenhuis.

**Writing – original draft:** Marieke H. Edwards.

**Writing – review & editing:** Marieke H. Edwards, Linda van den Dries, Sten-Åke Stenberg, Judith R. L. M. Wolf.

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
