## [Decision Letter · Decision Letter 0]

25 Nov 2020

PONE-D-20-27239

Dutch tenants at risk of eviction: Identifying predictors of eviction orders

PLOS ONE

Dear Dr. Edwards,

Thank you for submitting your manuscript to PLOS ONE. After careful consideration, we feel that it has merit but does not fully meet PLOS ONE’s publication criteria as it currently stands. Therefore, we invite you to submit a revised version of the manuscript that addresses the points raised during the review process.

Both qualified reviewers liked your research but raised valid concerns. Please try to address their concerns as much as you can. Particularly, you have not explored enough your data. Although it is difficult to establish a rigorous causality test, it should be feasible to estimate more models with different predictors to see how your current findings change, as reviewer 2 suggested. Also a p value of 0.1 is often considered significant too and you certainly can include those variables whose coefficients are statistically significant at the 10% level or better.  

We look forward to receiving your revised manuscript.

Kind regards,

Shihe Fu, Ph.D.

Academic Editor

PLOS ONE

Journal Requirements:

2. Please ensure you have provide a copy of all questionnaires used in your study as supporting information files.

Reviewers' comments:

Reviewer's Responses to Questions

**Comments to the Author**

1. Is the manuscript technically sound, and do the data support the conclusions?

Reviewer #1: Partly

Reviewer #2: Partly

2. Has the statistical analysis been performed appropriately and rigorously? 

Reviewer #1: No

Reviewer #2: Yes

3. Have the authors made all data underlying the findings in their manuscript fully available?

Reviewer #1: No

Reviewer #2: Yes

4. Is the manuscript presented in an intelligible fashion and written in standard English?

Reviewer #1: Yes

Reviewer #2: Yes

5. Review Comments to the Author

Reviewer #1: This paper utilizes a sample of 344 tenants who were at risk of eviction due to rent arrears in five Dutch cities interviewed using a structured questionnaire to test ten potential risk factors for evictions. Multiple logistic regression analyses are applied on the tenants’ personal information and the tenants’ situation at the notice of eviction risk and six months after the notice. The authors conclude that the amount of rent arrears was a strong predictor for receiving an eviction order, and single tenants, tenants who had already been summoned to appear in court were more likely to receive an eviction order. The authors believe that these results can contribute to identifying households at risk of eviction at an early stage, and help to develop targeted interventions to prevent evictions.

Overall, this is an interesting research work and carries immediate real-life application potentials. However, there are several major weaknesses in the current version of this paper.

First, the survey design. The sample size is not big, but this is not a very big issue. The more important issue is the representativeness of the samples. The authors need to provide more justifications on the interview was scientifically designed, and the interviewees were not chosen with systematic bias. For example, since the interviewees enrolled in the survey on voluntary basis rather than by force, there is high possibility that only those tenants with low self-assessment of eviction risks would choose to join the research. Thus, the analysis on such groups of interviewees may produce biased findings.

Second, the empirical design. What so far observed and discussed by the authors is just correlation but not causality. Correlation helps little in delivering policy recommendation. Since the data is one-shot cross-section data, it may suffer high risks of self-selection and endogeneity when attempting to discover causality relationship. These problems are hard to address but not incurable. There are several econometric tools to address them. The authors may carefully consider them.

Third, the model. Too few controls are included in the regression models. I suppose several variables such as ethnic backgrounds, immigrant or not and years of immigration, the ratio of rent in household monthly income, and regional dummies should enter in the model. The work also needs several robustness checks, i.e., checking the consistency of the findings in different model setup and alternative use of key variables.

Fourth, the generality. PLOS ONE is an international journal, the authors should put more justifications on why a case study of Dutch may be interested to international audiences, or what international readers can learn from this work.

Finally, contributions and innovations. The authors should make more attempts on highlighting what new insights this paper could offer to readers. The findings such as “the amount of rent arrears was a strong predictor for receiving an eviction order”, is well predicted and not surprised at all. The readers need to know something that they could not imagine themselves but need to be informed from researchers.

Reviewer #2: The authors have a fine idea to explore some potential risk factors predicting whether or not tenants receive an eviction order. In particular, they use the logistic regression model to determine which factor (variable) significantly affects the probability of tenants receive an eviction order. The authors conclude that the amount of rent arrears, whether tenants are single, and phase in eviction process are strong predictors for eviction orders. While the topic is of interest, I do have some concerns that should be addressed and discussed in the paper.

1. While 495 tenants were included in the study at T0, only 71% (353) of tenants were tracked at T1, which may cause a selection bias issue. For example, as you mentioned in line 294-296 (Page 13), tenants who have had a court hearing may not want to continue to participate in an interview. Therefore, I suggest that you can use the Probit or Logit model to investigate which factors affect the response rates for follow-up interview at T1. In addition, you can also test whether significant differences exist between the respondents with and without information about eviction for the ten predictors.

2. You conducted a backward stepwise logistic regression analysis to discard the statistically insignificant variables, one by one. However, I am afraid that the results based on the model you used may be misleading. Smith (2018) uses a Monte Carlo simulation model to demonstrate that some predictors that have causal effects on the dependent variable may happen be statistically insignificant and therefore are discarded, while nuisance predictors may be coincidentally significant. Plus, the larger the number of potential predictors, the stepwise regression analysis is less effective.

From the perspective of econometrics, variables should be included in a model based on theoretical grounds rather than the size of their t-values. If variables that truly belong in the model are omitted, the estimated coefficients of the existing explanatory variables would be biased. (Wooldridge 2006)

Based on the reasons mentioned above, I suggest that you can choose variables from a priori knowledge, then use the logistic regression model to determine which variables are significant and which are not.

3. When you try to explain the results shown in Table 2, you mainly show that how the odds ratios (OR) of receiving an eviction order change when predictors (such as household composition) change. I think that you can also show the marginal effects of receiving an eviction order at the means for all available predictors, because a marginal effect of a predictor shows how do predicted probabilities change as the predictor changes 1-unit, which would be easier to understand for audience.

4. You can try to use other functional forms when you run the logistic regression. For example, I suggest that you can transform some variables such as the total household income in the last month and the total amount of respondents’ current rent arrears using the natural logarithm transformation.

5. You can try to do subsample regression to explore heterogeneity among different groups. For example, you can divide the whole sample into group of one-person household and group of multi-person household.

Some more detailed comments while reading the paper:

-Figure 1: You’d better use a vector image (such as SVG, eps, wmf，emf), because vector images can be enlarged to any size without appearing pixelated.

-Page 10: The last sentence (line 226-228) on Page 10 is unclear to me. “The final model was used to calculate the risk of 227 receiving an eviction order as a result of combinations of two predictors, controlling for the 228 other predictors in the models.” Could you explicitly point out the names of the two predictors you mentioned?

-Figure 2 and 3: Please show the confidence interval of variables in Figure 2 and 3.

References:

Smith, Gary. "Step away from stepwise." Journal of Big Data 5.1 (2018): 32.

Wooldridge JW. Introductory econometrics: a modern approach. 3rd ed. Mason: Thompson; 2006. p. 94–7.

6. PLOS authors have the option to publish the peer review history of their article (what does this mean?). If published, this will include your full peer review and any attached files.

Reviewer #1: No

Reviewer #2: No

---

## [Author Response · Author response to Decision Letter 0]

17 May 2021

Thank you for providing us an opportunity to revise our manuscript “Dutch Tenants at Risk of Eviction: Identifying Predictors of Eviction Orders”. In this letter, we respond to each point raised. Please note: page and line numbers in our responses refer to the document titled “Revised Manuscript with Track Changes”.

Response to the Editor:

The Editor indicated the we had not explored our data enough, and that more models with different predictors should be estimated. The Editor also suggested using a p value of 0.1 instead of 0.05.

We repeated the backward stepwise logistic regression analysis using a p value of 0.1; this resulted in the same model. We also estimated models with the other predictors (see our response to reviewer #2 below).

Response to Reviewer #1:

Reviewer #1 raised five issues, related to the survey design, the empirical design, the model, the generality, and contributions and innovations.

Reviewer #1: First, the survey design. The sample size is not big, but this is not a very big issue. The more important issue is the representativeness of the samples. The authors need to provide more justifications on the interview was scientifically designed, and the interviewees were not chosen with systematic bias. For example, since the interviewees enrolled in the survey on voluntary basis rather than by force, there is high possibility that only those tenants with low self-assessment of eviction risks would choose to join the research. Thus, the analysis on such groups of interviewees may produce biased findings.

Due to moral and ethical constraints the medical ethical committee of the region Arnhem-Nijmegen allowed for inclusion of tenants on a voluntary basis. We cannot force tenants to participate in our study. This indeed means there is a risk of an inclusion bias with a higher chance of tenants with a lower self-assessment of eviction risks joining the study. The results of our study, however, show that tenants participating in the study experienced high levels of stress, shame, sadness, powerlessness, and trouble sleeping, related to the rent arrears. This suggests that tenants with a high self-assessment of eviction risk also entered the study. We added sections about this in the Methods and Discussion (page 8, lines 173-181; page 20, lines 438-442).

Reviewer #1: Second, the empirical design. What so far observed and discussed by the authors is just correlation but not causality. Correlation helps little in delivering policy recommendation. Since the data is one-shot cross-section data, it may suffer high risks of self-selection and endogeneity when attempting to discover causality relationship. These problems are hard to address but not incurable. There are several econometric tools to address them. The authors may carefully consider them.

Thank you for this feedback. However, our data is not only “one-shot cross-section data”; we collected data on predictors among tenants at risk of eviction, and six months later we collected the outcome data. So, besides correlation, there is also a temporal priority of the independent variables. The T1 data collection is described on page 7, lines 153-159. We edited this paragraph to improve clarity.

Reviewer #1: Third, the model. Too few controls are included in the regression models. I suppose several variables such as ethnic backgrounds, immigrant or not and years of immigration, the ratio of rent in household monthly income, and regional dummies should enter in the model. The work also needs several robustness checks, i.e., checking the consistency of the findings in different model setup and alternative use of key variables.

As suggested, we added the regression model with all predictors (page 12, lines 241-250), presenting an overview of the controls that we tested in the model. These predictors were all derived from our review of the literature on risk factors for eviction. The backward stepwise logistic regression analysis was followed by forward stepwise logistic regression analysis, adding one predictor to the sparse model at a time to determine if that improved the model.

We also conducted the logistic regression with the natural logarithm transformations for household income and rent arrears; however, this did not change the results. Therefore, we chose not to report on the results using these transformations. Additionally, we repeated the analyses for single tenants versus multi-person households, which led to two different models (pages 14- 16, lines 305-352). 

Reviewer #1: Fourth, the generality. PLOS ONE is an international journal, the authors should put more justifications on why a case study of Dutch may be interested to international audiences, or what international readers can learn from this work.

We elaborated more on the generalizability of the study (page 19, lines 417-430). We also hope that the addition of the comparison between single tenants and multi-person households (page 18, lines 390-400) will be more interesting to international audiences. We do realize that local laws, policies and regulations around rent arrears and evictions vary greatly, and we therefore highly recommend further research in other locations to determine whether the relations we found are present across various localities (page 21, lines 468-469).

Reviewer #1: Finally, contributions and innovations. The authors should make more attempts on highlighting what new insights this paper could offer to readers. The findings such as “the amount of rent arrears was a strong predictor for receiving an eviction order”, is well predicted and not surprised at all. The readers need to know something that they could not imagine themselves but need to be informed from researchers.

As suggested by Reviewer #2, we conducted further analyses and compared single tenants and multi-person households. These analyses resulted in more insights (the role of job loss and differences between cities; page 18, lines 390-400). We hope that this addition resolves this issue.

Response to Reviewer #2:

Reviewer #2 raised five issues. We will respond to each issue below.

Reviewer #2: While 495 tenants were included in the study at T0, only 71% (353) of tenants were tracked at T1, which may cause a selection bias issue. For example, as you mentioned in line 294-296 (Page 13), tenants who have had a court hearing may not want to continue to participate in an interview. Therefore, I suggest that you can use the Probit or Logit model to investigate which factors affect the response rates for follow-up interview at T1. In addition, you can also test whether significant differences exist between the respondents with and without information about eviction for the ten predictors.

The T1 data was not collected through interviews with tenants, but by asking housing associations to provide the needed information about each tenant. Because the issues preventing us from collecting complete T1 data were at the housing association level (organizational restructuring, changes in IT systems and difficulties working with debt collectors) and not at the individual tenant level, we believe this has not caused a selection bias at T1. The issues around the T1 data collection are described under “Procedure and Participants” (pages 7-8, lines 159-172). We added the discussion of the T1 data collection in the Discussion (page 20, lines 443-448) to be clearer.

Reviewer #2: You conducted a backward stepwise logistic regression analysis to discard the statistically insignificant variables, one by one. However, I am afraid that the results based on the model you used may be misleading. Smith (2018) uses a Monte Carlo simulation model to demonstrate that some predictors that have causal effects on the dependent variable may happen be statistically insignificant and therefore are discarded, while nuisance predictors may be coincidentally significant. Plus, the larger the number of potential predictors, the stepwise regression analysis is less effective.

From the perspective of econometrics, variables should be included in a model based on theoretical grounds rather than the size of their t-values. If variables that truly belong in the model are omitted, the estimated coefficients of the existing explanatory variables would be biased. (Wooldridge 2006)

Based on the reasons mentioned above, I suggest that you can choose variables from a priori knowledge, then use the logistic regression model to determine which variables are significant and which are not.

We understand the concerns about the risk of omitting variables that should be in the model. Therefore, we added a section in the results (page 12, lines 241-250) that presents the model that includes all variables that we expected to have an effect on receiving an eviction order. These variables were all derived from the literature presented in the introduction. This is stated in the Method section (page 8, line 184). Additionally, after determining the sparse model with only significant variables, we added each excluded variable to this sparse model again, to determine if adding the variable would improve the model. We clarified this in the Results in more detail (page 12, lines 253-255)

Reviewer #2: When you try to explain the results shown in Table 2, you mainly show that how the odds ratios (OR) of receiving an eviction order change when predictors (such as household composition) change. I think that you can also show the marginal effects of receiving an eviction order at the means for all available predictors, because a marginal effect of a predictor shows how do predicted probabilities change as the predictor changes 1-unit, which would be easier to understand for audience.

Thank you for this feedback. We added the marginal effects (page 13, lines 270-273 and lines 276-280).

Reviewer #2: You can try to use other functional forms when you run the logistic regression. For example, I suggest that you can transform some variables such as the total household income in the last month and the total amount of respondents’ current rent arrears using the natural logarithm transformation.

Thank you for this suggestion. We ran the logistic regression with the natural logarithm transformations for household income and rent arrears; however, this did not change the results. Therefore, we chose not to report on the results using these transformations.

Reviewer #2: You can try to do subsample regression to explore heterogeneity among different groups. For example, you can divide the whole sample into group of one-person household and group of multi-person household.

This is a great suggestion, thank you. We conducted the logistic regression analysis among single tenants and multi-person households and found interesting differences between these groups (pages 14-16, lines 305-352). We explored options to compare other groups, but because in most cases the groups were too small to be able to conduct the analyses, we chose to only report on the difference between single tenants and multi-person households.

Reviewer #2: Figure 1: You’d better use a vector image (such as SVG, eps, wmf,emf), because vector images can be enlarged to any size without appearing pixelated.

Thank you for this suggestion. We uploaded Figure 1 as an EPS file instead.

Reviewer #2: Page 10: The last sentence (line 226-228) on Page 10 is unclear to me. “The final model was used to calculate the risk of receiving an eviction order as a result of combinations of two predictors, controlling for the other predictors in the models.” Could you explicitly point out the names of the two predictors you mentioned?

We changed the language of this section (pages 11-12, lines 237-238). In the Results section (page 14, lines 281-285) we explain exactly which predictors were included in these risk calculations.

Reviewer #2: Figure 2 and 3: Please show the confidence interval of variables in Figure 2 and 3.

Because Figures 2 and 3 were created by computing the probability of receiving an eviction order based on the final logistic regression model, and the graphs were not based on the data but on the parameters in the model, we cannot include confidence intervals.

In addition to the changes requested by the Reviewers and Editor, we also edited the acknowledgement section, based on communication with your editorial office regarding one author who passed away before completion of the manuscript. Additionally, we uploaded the questionnaire as supplemental material.

---

## [Decision Letter · Decision Letter 1]

29 Jun 2021

Dutch tenants at risk of eviction: Identifying predictors of eviction orders

PONE-D-20-27239R1

Dear Dr. Edwards,

We’re pleased to inform you that your manuscript has been judged scientifically suitable for publication and will be formally accepted for publication once it meets all outstanding technical requirements.

Kind regards,

Shihe Fu, Ph.D.

Academic Editor

PLOS ONE

P.S. I am very sorry for your loss of your coauthor Professor Manfred te Grotenhuis.  Please accept my deepest condolences.

Reviewers' comments:

Reviewer's Responses to Questions

**Comments to the Author**

1. If the authors have adequately addressed your comments raised in a previous round of review and you feel that this manuscript is now acceptable for publication, you may indicate that here to bypass the “Comments to the Author” section, enter your conflict of interest statement in the “Confidential to Editor” section, and submit your "Accept" recommendation.

Reviewer #1: All comments have been addressed

Reviewer #2: All comments have been addressed

2. Is the manuscript technically sound, and do the data support the conclusions?

Reviewer #1: Yes

Reviewer #2: Yes

3. Has the statistical analysis been performed appropriately and rigorously? 

Reviewer #1: Yes

Reviewer #2: Yes

4. Have the authors made all data underlying the findings in their manuscript fully available?

Reviewer #1: Yes

Reviewer #2: Yes

5. Is the manuscript presented in an intelligible fashion and written in standard English?

Reviewer #1: Yes

Reviewer #2: Yes

6. Review Comments to the Author

Reviewer #1: My concerns have been addressed and I have no further comments. But I would like to receive information on how other reviewers judge the revision.

Reviewer #2: The authors have addressed all the issues I raised. Therefore I accept this paper without further comments. Congratulations.

7. PLOS authors have the option to publish the peer review history of their article (what does this mean?). If published, this will include your full peer review and any attached files.

Reviewer #1: No

Reviewer #2: **Yes: **ZHEJIN ZHAO

---

## [Editor Report · Acceptance letter]

2 Jul 2021

PONE-D-20-27239R1 

Dutch tenants at risk of eviction: Identifying predictors of eviction orders 

Dear Dr. Edwards:

I'm pleased to inform you that your manuscript has been deemed suitable for publication in PLOS ONE. Congratulations! Your manuscript is now with our production department. 

Kind regards, 

on behalf of

Dr. Shihe Fu 

Academic Editor

PLOS ONE